# Repurposing FDA Drug Compounds against Breast Cancer by Targeting EGFR/HER2

**DOI:** 10.3390/ph14080791

**Published:** 2021-08-12

**Authors:** Irving Balbuena-Rebolledo, Itzia Irene Padilla-Martínez, Martha Cecilia Rosales-Hernández, Martiniano Bello

**Affiliations:** 1Laboratorio de Diseño y Desarrollo de Nuevos Fármacos e Innovación Biotecnológica, Escuela Superior de Medicina, Instituto Politécnico Nacional, Plan de San Luis y Diaz Mirón, s/n, Col. Casco de Santo Tomas, Ciudad de México 11340, Mexico; irving.balbu@gmail.com; 2Laboratorio de Química Supramolecular y Nanociencias, Unidad Profesional Interdisciplinaria de Biotecnología, Instituto Politécnico Nacional, Av. Acueducto s/n, Barrio La Laguna Ticomán, Ciudad de México 07340, Mexico; ipadillamar@ipn.mx; 3Laboratorio de Biofísica y Biocatálisis, Sección de Estudios de Posgrado e Investigación, Escuela Superior de Medicina, Instituto Politécnico Nacional, Plan de San Luis y Diaz Mirón, s/n, Col. Casco de Santo Tomas, Ciudad de México 11340, Mexico; marcrh2002@yahoo.com.mx

**Keywords:** HER2, EGFR, doxazosin, docking, MD simulations

## Abstract

Repurposing studies have identified several FDA-approved compounds as potential inhibitors of the intracellular domain of epidermal growth factor receptor 1 (EGFR) and human epidermal receptor 2 (HER2). EGFR and HER2 represent important targets for the design of new drugs against different types of cancer, and recently, differences in affinity depending on active or inactive states of EGFR or HER2 have been identified. In this study, we first identified FDA-approved compounds with similar structures in the DrugBank to lapatinib and gefitinib, two known inhibitors of EGFR and HER2. The selected compounds were submitted to docking and molecular dynamics MD simulations with the molecular mechanics generalized Born surface area approach to discover the conformational and thermodynamic basis for the recognition of these compounds on EGFR and HER2. These theoretical studies showed that compounds reached the ligand-binding site of EGFR and HER2, and some of the repurposed compounds did not interact with residues involved in drug resistance. An in vitro assay performed on two different breast cancer cell lines, MCF-7, and MDA-MB-23, showed growth inhibitory activity for these repurposed compounds on tumorigenic cells at micromolar concentrations. These repurposed compounds open up the possibility of generating new anticancer treatments by targeting HER2 and EGFR.

## 1. Introduction

Human epidermal receptor type 2 (HER2) is part of a superfamily of human epidermal grown factor (EGF) receptors that include four receptors: ErbB1 (HER1 or EGFR), ErbB2 (HER2), ErbB3 (HER3), and ErbB4 [1]. Of these receptors, EGFR and HER2 are the most significant targets for anticancer therapy. These receptors are composed of a transmembrane region, extracellular domain, and receptor tyrosine kinase (RTK) domain, and the latter has been one of the major targets for anticancer drug design [2]. Activation of these receptors begins with the binding of endogenous growth factors at the extracellular region, which encourages homo and heterodimerization of RTK of both receptors [3,4]. This produces a structural change in the activation loop of the RTK domain and a change in the N-lobe α-C helix to create active conformation [5,6,7,8]. The transition from the inactive to the active state is a two-step process for EGFR [5,6,7], whereas inactive, intermediate, and active states have been reported for HER2 [9,10,11]. Active and inactive EGFR conformations are used in drug design, identifying high specificity ligands by targeting the inactive conformation, whereas the active state is only promising when mutations promote the activated state [12]. Mutations, radiotherapy, and chemotherapy resistance give rise to the constitutive activation of both receptors, which is linked with the progression of different kinds of cancer, such as breast and lung cancers [13,14,15,16,17]. Food and Drug Administration (FDA)-drug kinase inhibitors are divided into two groups: those targeting the active and inactive EGFR states [18,19,20,21,22]. Of these, lapatinib and gefitinib are shown to be dual inhibitors of the inactive state of EGFR and HER2 [2,10,23,24,25,26,27]. Lapatinib and gefitinib are competitive inhibitors of ATP and are approved to treat colorectal, lung, pancreatic, and breast cancers [28,29]. However, the employment of these compounds is associated with drug resistance [30]. Therefore, it is necessary to find new EGFR and HER2 inhibitors. Recently, new EGFR inhibitors with similar structures to lapatinib were investigated [31,32,33,34,35]. Given that, we screened DrugBank in the quest for new FDA-approved compounds with structures similar to lapatinib and gefitinib. The twenty-four identified compounds were submitted to docking and molecular dynamics (MD) simulations with the molecular mechanics generalized Born surface area approach (MMGBSA) considering the inactive state of EGFR and HER2. In addition, calculations of the binding free energy values and per-residue free energy were determined to check the affinities of each ligand for both receptors. The repurposed compounds with the highest affinity for EGFR and HER2 showed growth inhibitory activity in two different breast cancer cell lines, MCF-7 and MDA-MB-231, at micromolar concentrations.

## 2. Results and Discussion

### 2.1. Docking Studies

Blind docking studies showed that alfuzosin, amodiaquine, antrafenine, bopindolol, carvedilol, doxazosin, irinotecan, pindolol, prazosin, quinacrine, reserpine, saprisartan, sildenafil, terazosin, topotecan, trimetrexate, and udenafil reached the binding site of HER2 with a binding score between −7.6 and −9.4 kcal, whereas deserpidine, rifaximin, vardenafil, vinblastine, vincristine, vindesine, and vinorelbine did not (Appendix A). The HER2-binding group of compounds, in addition to deserpidine and vardenafil, exhibited affinity to EGFR that ranged between −6.9 and −9.1 kcal mol^−1^ (Appendix A). However, a lack of affinity of rifaximin, vinblastine, vincristine, vindesine, and vinorelbine for both EGFR and HER2 was observed. All HER2-ligand complexes were stabilized by between 17 and 27 residues at the HER2 binding site, from which the following residues, L726, F731, V734, A751, I752, K753, M774, S783, L785, L796, V797, T798, Q799, L800, M801, Y803, G804, C805, L852, T862, D863, and F864, were mostly present (Appendix A). EGFR-ligand complexes were also stabilized by between 17 and 27 residues at the EGFR binding site; of these residues, L718, V726, A743, K745, T790, Q791, L792, M793, G796, C797, D800, R841, L844, T854, D855, F856, and L858 were present in most of the complexes (Appendix A). Both HER2- and EGFR-ligand complexes were further submitted to MD simulation to evaluate the prevalence of the interactions predicted by docking studies.

### 2.2. Convergence and Equilibrium

Complexes between HER2 and alfuzosin, antrafenine, bopindolol, carvedilol, doxazosin, irinotecan, pindolol, prazosin, quinacrine, saprisartan, sildenafil, terazosin, topotecan, and trimetrexate showed stable protein-ligand complexes during simulation, whereas amodiaquine, reserpine, and udenafil dissociated from the HER2 receptor during simulation; therefore, these three compounds were discarded from the analysis. For EGFR, stable protein-ligand complexes with alfuzosin, amodiaquine, antrafenine, bopindolol, carvedilol, doxazosin, pindolol, prazosin, quinacrine, saprisartan, terazosin, topotecan, trimetrexate, udenafil, and vardenafil were observed, whereas deserpidine, irinotecan, reserpine, and sildenafil diffused from the EGFR binding site through simulation, therefore these four compounds were discarded from the study. The RMSD analysis showed that the HER2-ligand and EGFR-ligand complexes reached constancy between 20 and 50 ns with RMSD values oscillating between 1.65 and 3.47 Å. The radius of gyration (RG) analysis revealed that these complexes also exhibited permanence from 20 to 50 ns with values fluctuating between 18.90 and 20.24 Å (Appendix A). Therefore, additional analyses were performed removing the first 50 ns.

### 2.3. Structural Analysis of HER2-Ligand Complexes

Clustering analysis over the equilibrated simulation time was performed to obtain the most populated conformation that allowed for exploration of the structural differences between compounds on EGFR and HER2. Analysis of the complexes between HER2 and alfuzosin, antrafenine, bopindolol, carvedilol, doxazosin, irinotecan, pindolol, prazosin, quinacrine, saprisartan, sildenafil, terazosin, topotecan, and trimetrexate showed that these complexes were stabilized by between 17 and 27 residues (Table 1). From these residues, L726, F731, V734, A751, I752, K753, M774, S783, R784, L785, L796, V797, T798, Q799, L800, M801, Y803, G804, C805, L852, T862, D863, and F864 were present in most of the complexes. Some of these residues participate in forming hydrogen bonds with compounds. Alfuzosin was stabilized by four hydrogen bonds with backbone atoms of M801 and F864 and through side chain atoms of S783 and D863 (Figure 1A). Bopindolol formed two hydrogen bonds with backbone atoms of L726 and C805 and one hydrogen bond with side chain atoms of D863 (Figure 1C). Carvedilol formed one hydrogen bond with side chain atoms of D863 (Figure 1D). Doxazosin forms two hydrogen bonds with backbone atoms of M801 and Y803 and two hydrogen bonds through side chain atoms of S783 and T798 (Figure 1E). Irinotecan formed two hydrogen bonds with backbone atoms L726 and Y803 and a Pi-Cation interaction with Phe864 (Figure 2A). Pindolol forms two hydrogen bonds with C805 and D808 through backbone and side chain atoms, respectively (Figure 2B).

Prazosin forms one hydrogen bond with the backbone atoms of M793 (Figure 2C). Saprisartan forms one hydrogen bond with backbone atoms of D863. Y803 and T798 form hydrogen bonds with sildenafil through backbone and side chain atoms, respectively (Figure 2E). The side chain atoms of S783, T798, and D863 form hydrogen bonds with terazosin, and F864 forms through backbone atoms (Figure 3B). Topotecan formed one hydrogen bond with side chain atoms of D808 (Figure 3C). Trimetrexate established three hydrogen bonds with backbone atoms of M801, Y803, and F864 and one hydrogen bond with side chain atoms of T862 (Figure 3D). Analysis of frequent residues revealed the presence of characteristic interactions with M801, which is found for other inhibitors of HER2 [36,37]. In addition, F864 and Y803 formed hydrogen bonds in five HER2-ligand complexes.

### 2.4. Structural Analysis of EGFR-Ligand Complexes

Analysis of the complexes between EGFR and alfuzosin, amodiaquine, antrafenine, bopindolol, carvedilol, doxazosin, pindolol, prazosin, quinacrine, saprisartan, terazosin, topotecan, trimetrexate, udenafil, and vardenafil showed that these complexes were stabilized by between 13 and 24 residues (Table 2). From these protein-ligand complexes, L718, G719, S720, V726, A743, K745, T790, L792, M793, G796, C797, R841, and L844 appear in most complexes. The formation of hydrogen bonds was observed in some complexes. Alfuzosin forms hydrogen bonds with backbone atoms of D855 and side chains of T790 and D855 (Figure 4A). Amodiaquine makes hydrogen bonds with backbone atoms of L792 and C797 and with side chain atoms of D800 (Figure 4B). Bopindolol forms hydrogen bonds with side chain atoms of N842 and D855 (Figure 4D). Carvedilol forms one hydrogen bond with side chain atoms of D837 and one Pi-Cation interaction with F723 (Figure 4E). Doxazosin established two hydrogen bonds with backbone atoms of M793 and P794 and through side chain atoms of D804 (Figure 5A).

Pindolol makes four hydrogen bonds with side chain atoms of K745, N842, T854, and D855 (Figure 5B). Prazosin forms one hydrogen bond with backbone atoms of M793 (Figure 5C). Quinacrine forms one hydrogen bond with the backbone atoms of M793 and a salt bridge with D800 (Figure 5D). Saprisartan forms two hydrogen bonds with side chain atoms of K745 and R841 (Figure 5E). Terazosin forms one hydrogen bond with the backbone atoms of C797 (Figure 6A). Topotecan forms one hydrogen bond with backbone atoms of M793 (Figure 6B). Trimetrexate establishes two hydrogen bonds with side chain atoms of K745 and D855 (Figure 6C). Udenafil forms two hydrogen bonds through the backbone and side chain atoms of M793 and D800, respectively (Figure 6D). Vardenafil establishes one hydrogen bond through side chain atoms of R841 (Figure 6E). Stabilization of ligands at the ligand-binding site of EGFR establishes interactions with T790, whose mutations are associated with EGFR drug resistance [38,39]. We also identified characteristic interactions with M793, which was previously observed for other inhibitors of EGFR [36,37].

### 2.5. Affinity of Compounds

The difference in binding affinity for all complexes was determined using the MMGBSA approach. All systems showed thermodynamically favorable binding free energy (Δ*G_bind_*) values, where nonpolar interactions established by van der Waals energy (Δ*E_vdw_*) and nonpolar desolvation (Δ*G_npol,sol_*) dominated the binding of the protein-ligand complexes. Comparative analysis between both systems indicated that all the compounds exhibited a more favorable affinity to HER2 than EGFR (Table 3). HER2-ligand complexes showed that irinotecan (−56.4 ± 5 kcal/mol), quinacrine (−54.9 ± 3 kcal/mol), alfuzosin (−51.9 ± 6 kcal/mol), and antrafenine (−51.1 ± 3 kcal/mol) showed the highest binding free energy for HER2. The affinity of irinotecan and quinacrine was higher than that reported between HER2 and lapatinib (−51 ± 4 kcal/mol) [10] and gefitinib (−26 ± 6.0) [21]. The affinity observed for alfuzosin and antrafenine was similar to that of lapatinib [10] but higher than that of gefitinib [21]. Vardenafil (−48.2 ± 7 kcal/mol), alfuzosin (−45.4 ± 4 kcal/mol), terazosin (−44.7 ± 6 kcal/mol), and prazosin (−40.2 ± 4 kcal/mol) exhibited the highest affinity for EGFR. These four compounds showed a slightly lower affinity for EGFR than lapatinib (−58 ± 5 kcal/mol) [13] but a higher affinity than that reported between EGFR and gefitinib (−39 ± 2 kcal/mol) [21]. These results suggest that quinacrine, alfuzosin, and antrafenine could act as dual inhibitors of HER2 and EGFR with higher selectivity to HER2, whereas irinotecan only exhibited selectivity to HER2. Alfuzosin, terazosin, and prazosin could act as dual inhibitors of EGFR and HER2, whereas vardenafil only exhibited selectivity for EGFR.

### 2.6. Per-Residue Decomposition Free Energy

This analysis allowed the identification of the residues that provided the most Δ*G_bind_* values for compounds that exhibited more energetically favorable Δ*G_bind_* values. Table 4 shows that L726, F731, V734, A751, K753, M774, S783, L785, L796, V797, T798, M801, C805, L852, T862, D863, and F864 were the main contributors to the stabilization of the HER2_alfuzosin_ complex, from which S783, M801, D863 and F864 established hydrogen bonds with polar atoms of alfuzosin (Figure 1A). In the HER2_antrafenine_ complex, Val726, G727, T733, V734, A751, K753, L785, L796, V797, T798, M801, C805, L852, T862, and D863 contributed the most to the maintenance of the complex (Figure 1B). In the HER2_irinotecan_ complex, Val726, G727, F731, V734, A751, K753, S783, L785, L796, V797, T798, P802, Y803, C805, L852, T862, and F864 supported the highest preservation of the complex, from which Y803 and Val726 formed hydrogen bonds, and F864 formed Pi-cation interactions with irinotecan (Figure 2A). In the HER2_quinacrine_ complex, Val726, F731, V734, A751, K753, S783, L785, T798, L800, P802, Y803, G804, C805, L852, and T862 contributed the most to the conservation of this complex (Figure 2D).

Table 5 shows that L718, V726, A743, K745, L788, I789, T790, M793, G796, C797, L844, T854, F856, and L858 contributed the most to the Δ*G_bind_* value of the EGFR_alfuzosin_ complex, and of these residues, T790 forms hydrogen bonds with alfuzosin (Figure 4A). In the EGFR_prazosin_ complex, L718, V726, A743, K745, L777, L788, T790, L792, M793, G796, C797, L844, T854, and F856 are the main contributors to the affinity of this complex, of which M793 established one hydrogen bond with prazosin (Figure 5C). In the EGFR_terazosin_ complex, L718, V726, A743, I744, K745, M766, L777, L788, I789, T790, L792, M793, G796, C797, L844, T854, and F856 are major sources of the stability of this complex, of which C797 formed one hydrogen bond with a polar atom of prazosin (Figure 6A). L718, G719, G721, F723, T725, V726, A743, K745, L792, M793, G796, C797, R841, L844, and L858 represent the main origin of the permanence of the EGFR_vardenafil_ complex, of which R841 established one hydrogen bond with vardenafil (Figure 6E).

### 2.7. Antiproliferative Assays

HER2 and EGFR are rare important watchdogs for normal cellular activities, and their dysregulation has been linked to protein overexpression that promotes the progression of several kinds of cancer [40,41]. MDA-MB-231 and MCF-7 cell lines are estrogen receptor-negative and positive, respectively [42], and both expressed EGFR and HER2, although MDA-MB-231 cells expressed HER2 and EGFR at higher concentrations than MCF-7 cells [43]. The evaluation of growth inhibition by alfuzosin, quinacrine, terazosin, prazosin, and irinotecan was conducted using the MTT assay in selected MDA-MB-231 (Figure 7) and MCF-7 (Figure 8) cell lines and compared with that of gefitinib and lapatinib. Although vardenafil and antrafenine exhibited good theoretical affinities for EGFR or HER2, they were not included in this study because they were not commercially available at the time this research was performed.

As shown in Table 6, all evaluated compounds exhibited IC_50_ values in the µM range. The compounds with the best antiproliferative activity in the MCF-7 cell line were irinotecan and quinacrine, which exhibited greater antiproliferative activity than gefitinib or lapatinib. In the MDA-MB-231 cell line, quinacrine exhibited better antiproliferative activity than lapatinib or gefitinib. Prazosin and irinotecan exhibited better antiproliferative activity than terazosin and alfuzosin but lower antiproliferative activity than lapatinib and gefitinib. Although the affinity tendency predicted by the MMGBSA approach was not in line with the antiproliferative activity, the results support the affinity observed through theoretical methods of alfuzosin, terazosin, prazosin, and irinotecan over HER2 and EGFR.

## 3. Methods

### 3.1. Preparation of Systems

Twenty-four FDA small drugs structurally similar to lapatinib and gefitinib, alfuzosin, amodiaquine, antrafenine, bopindolol, carvedilol, deserpidine, doxazosin, irinotecan, pindolol, prazosin, quinacrine, rifaximin, sildenafil, reserpine, saprisartan, terazosin, topotecan, trimetrexate, udenafil, vardenafil, vinblastine, vincristine, vindesine, and vinorelbine (Appendix A), were taken from the DrugBank [44] by retrieving the lapatinib and gefitinib charts via the option: show similar structures for approved drugs. These compounds were optimized at the AM1 level with Gaussian 09 W software [45]. The inactive state of EGFR was retrieved from the protein data bank (PDB code 1XKK). The inactive state of HER2 was selected from a previous study [10].

### 3.2. Docking Studies

The twenty-four FDA small compounds were docked with EGFR and HER2 with AutoDock Tools 1.5.6 and AutoDock 4.2 programs [46]. Hydrogen atoms were included in ligands and receptors, and partial charges were assigned to receptors (Kollman) and ligands (Gasteiger). A grid size of 70 × 70 × 70 Å and 0.370 Å spacing was built on the receptor. A Lamarckian genetic algorithm was selected to evaluate a global conformational examination with a maximum of 1 × 10^7^ energy calculations and 200 separate populations. For each compound, 20 runs were calculated, and the best binding structures were chosen using the criteria of the lowest energetic ligand conformations at the binding site of the receptor.

### 3.3. Molecular Dynamics Simulations

Docking complexes were examined via MD simulation studies employing the AMBER16 package [47] and the ff14SB force field [48]. Systems were simulated in a dodecahedric box of 12 Å and solvated with the TIP3P water model [49]. Systems were neutralized with NaCl (0.15 M) to establish physiological strength. Ligand forcefields were constructed considering AM1-BCC atomic charges with the general Amber force field [50]. Solvated and neutralized systems were minimized using steepest descent through 4000 steps and equilibrated through 1 nanosecond (ns). Minimized and equilibrated systems were run by MD simulations for 100 ns with an NPT ensemble at 310 K, with each MD simulation run in triplicate. The time step for simulations was set to 2.0 fs. The SHAKE algorithm [51] was selected to restrict bonds at their equilibrium values. The PME method [52] was employed to describe the electrostatic term, and a 10 Å cutoff was chosen for the van der Waals forces. Constant temperature and pressure of 310 K and 1 atm, respectively, were kept using a weak-coupling algorithm [53], with coupling constants τ_T_ and τ_P_ of 1.0 and 0.2 ps, respectively. The results of the MD simulations were evaluated with the cpptraj tool in Amber16 to determine the root mean squared deviation (RMSD), the radius of gyration (R_G_), and clustering analysis. Figures were constructed using PyMOL [54] and Maestro Schrödinger version 10.5 [55].

### 3.4. Affinity Prediction and Per-Residue Decomposition

Binding free energy (Δ*G_bind_*) values and per residue energetic contribution were calculated using the MMGBSA approach [56,57,58,59]. Calculations were performed considering 500 protein-ligand complexes at intervals of 100 ps (last 50 ns of simulation) using implicit solvent models [60], a salt concentration of 0.10 M. Δ*G_bind_*_,_ and per-residue contributions for each protein-ligand complex were determined as previously described [11] and correspond to the median result of triplicate trials.

### 3.5. Biological Assays on Cell Lines

Gefitinib, lapatinib, terazosin, alfuzosin, prazosin, irinotecan, and quinacrine were purchased from Sigma Chemical (St. Louis, MO, USA). The breast cancer cell lines used in this study, MCF-7 and MDA-MB-231, were obtained from the American Type Tissue Culture Collection (ATCC), Rockville, MD, USA. MCF-7 and MDA-MB-231 cells were grown in Dulbecco’s modified Eagle’s medium (DMEM) high glucose with phenol red, and the culture medium was supplemented with 10% fetal bovine serum (FBS) (BioWest, Riverside, MO, USA) and 1% penicillin/streptomycin as an antibiotic. The cells were incubated in culture flasks (75 cm^3^) at 37 °C in a humidified atmosphere of 5% CO_2_ and 95% air, all of which were carried out under sterile conditions using a laminar flow hood.

### 3.6. Antiproliferative Assays on Cell Cultures

Breast cancer cell lines were detached with trypsin-EDTA (1%) for 5 min at 37 °C. After trypsin-EDTA inactivation with DMEM, the cells were centrifuged and resuspended in 4 mL of medium. Afterwards, cells were counted with CytoSmart cell counting (CytoSMART Technologies, Eindhoven, The Netherlands). Each cell line was seeded at 5 × 10^3^ cells in 96-well tissue culture plates and allowed to attach overnight (24 h) in a CO_2_ incubator before the assays. Cells were treated with the compounds at different concentrations, all compounds were dissolved in DMSO (0.1% final concentration), and the control contained medium with DMSO (0.1%). The assays were performed in triplicate for each concentration and incubated at 37 °C under a humid atmosphere with 5% CO_2_ for 48 h.

### 3.7. Cell Viability Assays

Further cell viability was determined using MTT [3-(4,5-dimethyl-2-thiazolyl)-2,5-diphenyl-2H-tetrazolium bromide (Sigma)]. After 48 h of incubation, the medium was removed and replaced by 100 µL of MTT (0.500 mg/mL), dissolved in PBS, and incubated for 3 h at 37 °C and 5% CO_2_. After that, the MTT/PBS was discarded, and 100 µL of DMSO was applied to each well to dissolve the dark-blue formazan crystals in intact cells. The resulting solution was measured by spectrophotometry with a microplate reader (Thermo Scientific, Multiskan^TM^ Sky) at a wavelength of 550 nm, and the quantity of formazan produced was directly proportional to the number of living cells. The results are expressed as the percentage of viable cells in relation to the control, whose viability was designated as 100%. Each data point was determined with *n* = 8 in three independent experiments, and the results are reported as the mean absorption ± SD using the GraphPad Prism 8 software.

## 4. Conclusions

We performed repurposing studies by screening DrugBank in the search for new FDA-approved drugs with chemical structures similar to lapatinib and gefitinib. Docking and MD simulations coupled to the MMGBSA approach using the selected DrugBank compounds, and considering the inactive state of EGFR and HER2, allowed us to identify that quinacrine, alfuzosin, and antrafenine could act as dual inhibitors of HER2 and EGFR but with higher selectivity to HER2, whereas irinotecan exhibited high selectivity to HER2. Alfuzosin, terazosin, and prazosin could act as dual inhibitors of EGFR and HER2, whereas vardenafil exhibited selectivity exclusively to EGFR. Per-residue decomposition analysis identified the main residues stabilizing the protein-ligand complexes with HER2 and EGFR systems, showing that V726, V734, A751, K753, L785, C805, L852, and T862 were present in the stabilization of HER2_alfuzosin_, HER2_antrafenine_, HER2_irinotecan_, and HER2_quinacrine._ In the stabilization of these complexes, the characteristic interaction with M801 in HER2_alfuzosin_ and HER2_antrafenine_ complexes was observed, which has been observed for other HER2 inhibitors were also observed. In the case of EGFR, L718, V726, A743, K745, M793, G796, C797, and L844 were present in the stabilization of EGFR_alfuzosin_, EGFR_prazosin_, EGFR_terazosin_, and EGFR_vardenafil_. All complexes formed interactions with M793, a characteristic interaction that has been observed for other inhibitors of EGFR. Finally, MTT assays showed that repurposed compounds exhibited antiproliferative activity on breast cancer cell lines, with irinotecan and quinacrine exhibiting greater antiproliferative activity than lapatinib and gefitinib in the MCF-7 cell line, and quinacrine exhibiting greater antiproliferative activity than lapatinib and gefitinib in the MCF-7 and MDA-MB-231 cell lines. Finally, we could show that our structure-based screening approach identifies novel repositioning candidates for the cancer target EGFR/HER2. Not only did we identify candidates structurally related to gefitinib and lapatinib, but also showed that they show the desired inhibitory activity on the target receptors. Particularly, the FDA-approved drugs irinotecan and quinacrine, which appeared as the top hit from our screen and were later validated, demonstrates the potential of our approach for drug repositioning.

## Figures and Tables

**Figure 1 pharmaceuticals-14-00791-f001:**
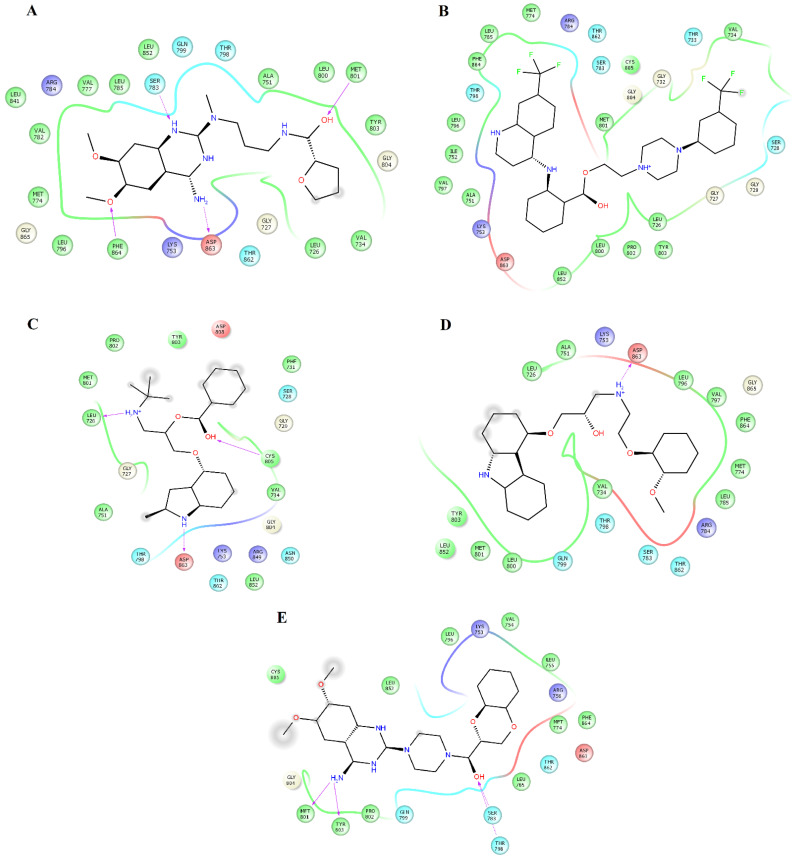
Map of contacts for the complexes between HER2 and alfuzosin, antrafenine, bipindolol, carvedilol, and doxazosin. Binding conformations for HER2_alfuzosin_ (**A**); HER2_antrafenine_ (**B**); HER2_bopindolol_ (**C**); HER2_carvedilol_ (**D**) and HER2_doxazosin_ (**E**). The figure was constructed with Maestro Schrödinger version 10.5.

**Figure 2 pharmaceuticals-14-00791-f002:**
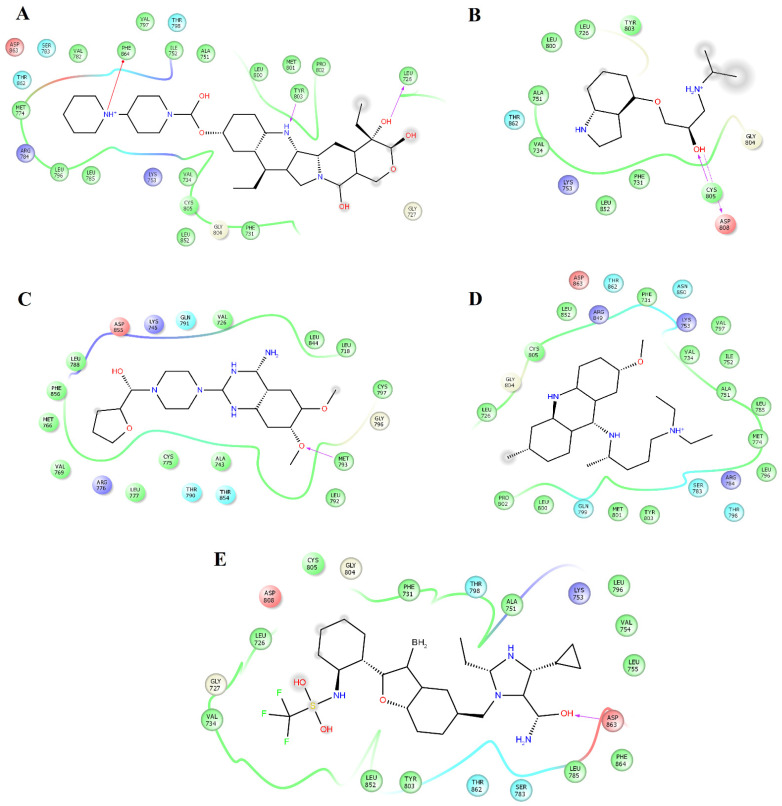
Map of interactions for the complexes between HER2 and irinotecan, pindolol, prazosin, quinacrine, and saprisartan. Binding conformations for HER2_irinotecan_ (**A**); HER2_pindolol_ (**B**); HER2_prazosin_ (**C**); HER2_quinacrine_ (**D**) and HER2_saprisartan_ (**E**).

**Figure 3 pharmaceuticals-14-00791-f003:**
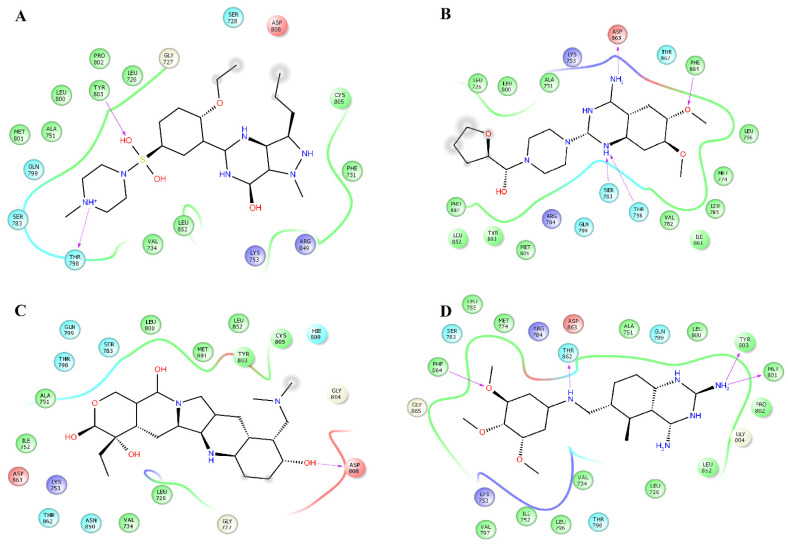
Map of interactions for the complexes between HER2 and sildenafil, terazosin, topotecan, and trimetrexate. Binding conformations for HER2_sildenafil_ (**A**); HER2_terazosin_ (**B**); HER2_topotecan_ (**C**) and HER2_trimetrexate_ (**D**).

**Figure 4 pharmaceuticals-14-00791-f004:**
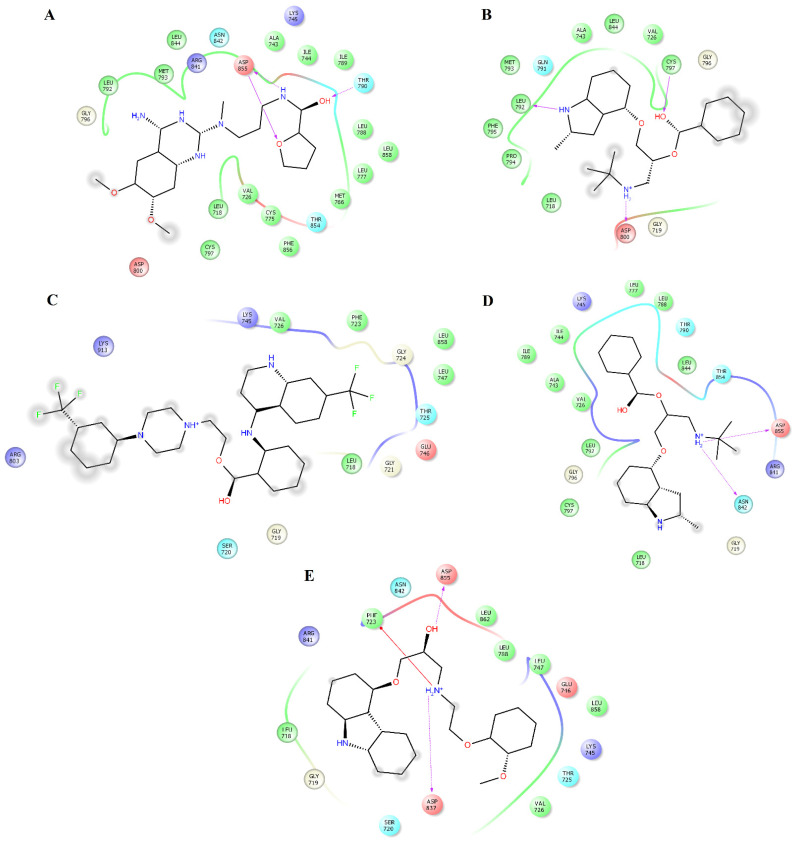
Interactions of the complexes between EGFR and alfuzosin, amodiaquine, antrafenine, bipindolol, and carvedilol. Binding complex for EGFR_alfuzosin_ (**A**); EGFR_amodiaquine_ (**B**); EGFR_antrafenine_ (**C**); EGFR_bipindolol_ (**D**) and EGFR_carvedilol_ (**E**).

**Figure 5 pharmaceuticals-14-00791-f005:**
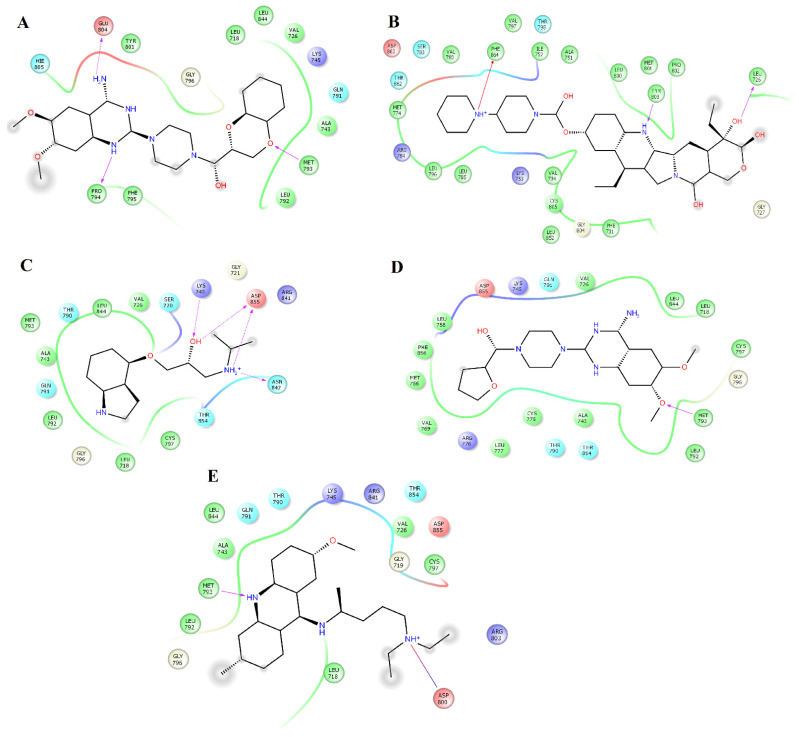
Interactions of the complexes between EGFR and doxazosin, pindolol, prazosin, quinacrine, and saprisartan. Binding complex for EGFR_doxazosin_ (**A**); EGFR_pindolol_ (**B**); EGFR_prazosin_ (**C**); EGFR_quinacrine_ (**D**) and EGFR_saprisartan_ (**E**).

**Figure 6 pharmaceuticals-14-00791-f006:**
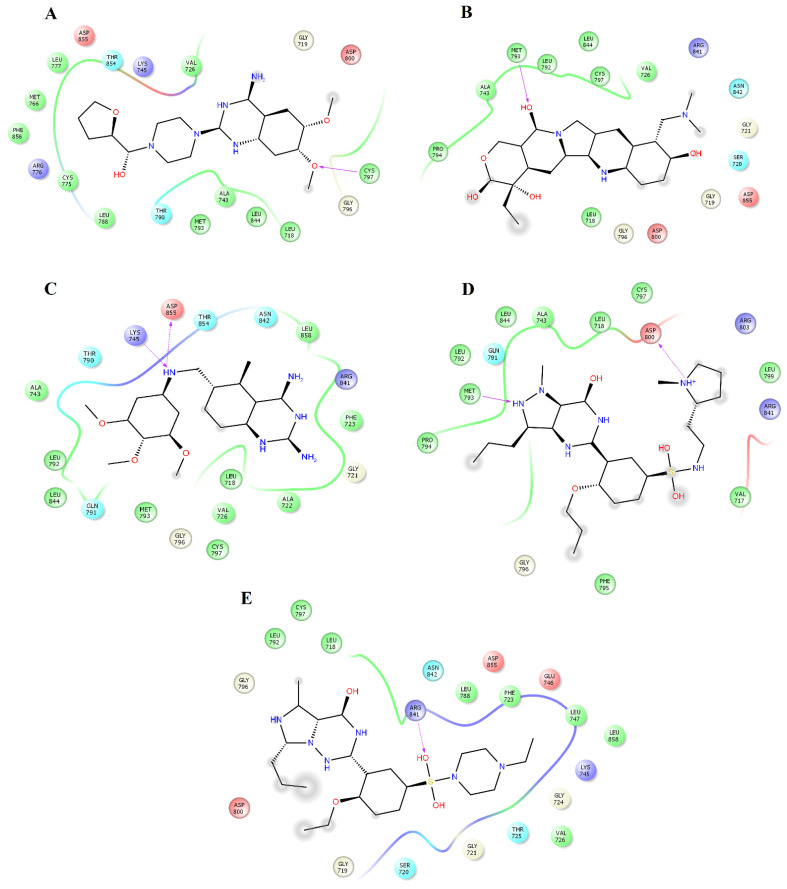
Interactions of the complexes between EGFR and terazosin, topotecan, trimetrexate, udenafil, and vardenafil. Binding complex for EGFR_terazosin_ (**A**); EGFR_topotecan_ (**B**); EGFR_trimetrexate_ (**C**); EGFR_udenafil_ (**D**) and EGFR_vardenafil_ (**E**).

**Figure 7 pharmaceuticals-14-00791-f007:**
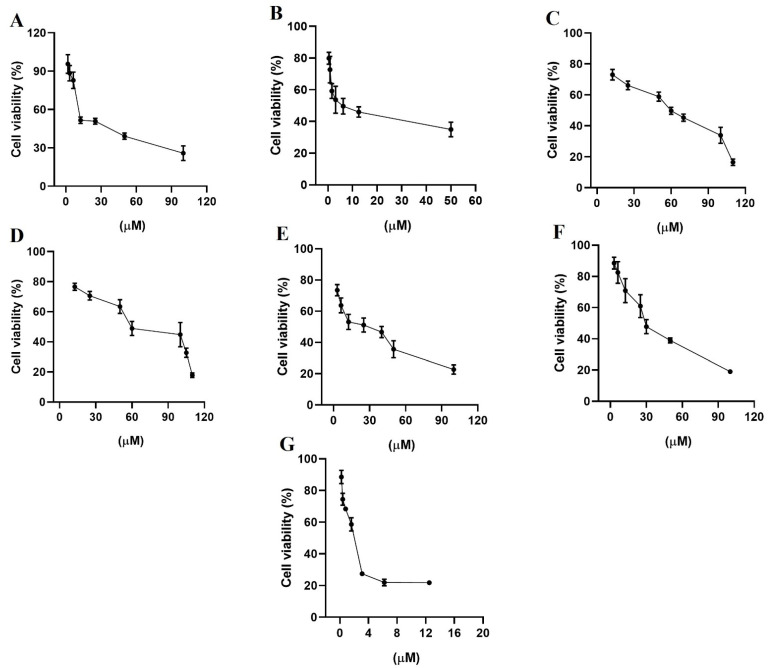
Percentage of cell viability from the biological study in the breast cancer cell line MDA-MB-231, (**A**) gefitinib; (**B**) lapatinib; (**C**) terazosin; (**D**) alfuzosin; (**E**) prazosin; (**F**) irinotecan; and (**G**) quinacrine. All assays were performed in triplicate.

**Figure 8 pharmaceuticals-14-00791-f008:**
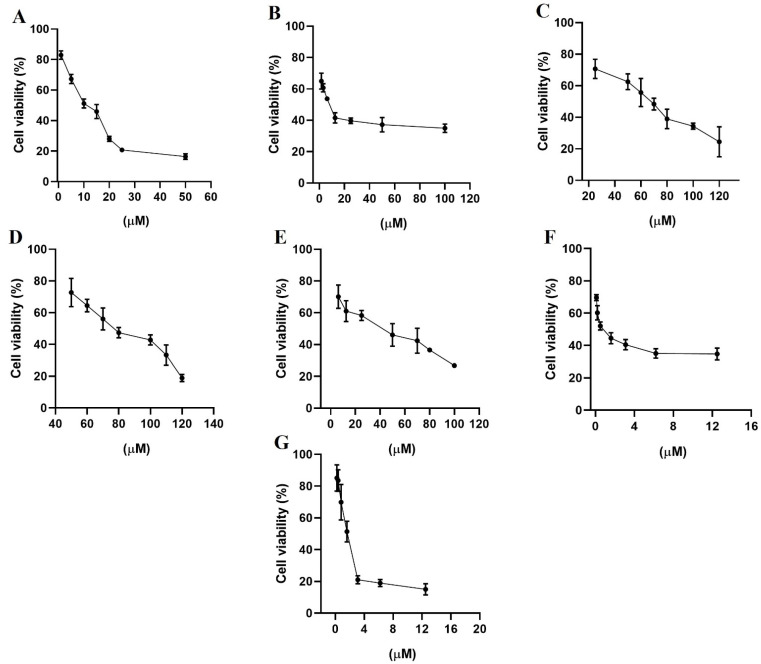
Percentage of cell viability from the biological study in the breast cancer cell line MCF-7, (**A**) gefitinib; (**B**) lapatinib; (**C**) terazosin; (**D**) alfuzosin; (**E**) prazosin; (**F**) irinotecan; and (**G**) quinacrine. All assays were performed in triplicate.

**Table 1 pharmaceuticals-14-00791-t001:** Interactions between inhibitors selected in DrugBank and HER2 during simulations.

Drug	Residues
Alfuzosin	L726, F731, V734, A751, I752, K753, M774, S783, R784, L785, L796, V797, T798, Q799, L800, M801, Y803, G804, C805, L852, T862, D863, F864.
Antrafenine	L726, F731, V734, A751, I752, K753, M774, S783, L785, L796, V797, T798, L800, M801, Y803, G804, C805, D808, H809, L852, V853, T862, D863, F864.
Bopindolol	L726, G727, F731, V734, A751, K753, S783, T798, Q799, L800, M801, P802, Y803, G804, C805, L852, T862, D863.
Carvedilol	L726, F731, V734, A751, K753, M774, S783, L785, L796, T798, Q799, L800, M801, Y803, G804, C805, L852, T862, D863, F864.
Doxasozin	V725, L726, S728, G727, K736, V734, K753, G804, C805, L807, D808, R849, N850, L852, T862, D863, L866
Irinotecan	L726, F731, V734, A751, K753, V754, L755, R756, I767, M774, S783, L785, L796, V797, T798, Q799, L800, M801, P802, Y803, G804, C805, L852, T862, D863, F864, G865.
Pindolol	L726, V734, A751, I752, K753, M774, S783, L785, L796, V797, T798, Q799, L800, M801, Y803, G804, L852, T862, D863, F864.
Prazosin	L726, F731, V734, A751, I752, K753, R756, M774, S783, R784, L785, L796, V797, T798, M801, Y803, G804, C805, L852, T862, D863, F864.
Quinacrine	L726, F731, V734, A751, I752, K753, M774, S783, L785, L796, V797, T798, Q799, L800, M801, Y803, G804, C805, R849, N850, L852, T862, D863, F864.
Saprisartan	L726, F731, V734, A751, I752, K753, V754, L755, R756, M774, S783, L785, L796, V797, T798, Q799, L800, M801, P802, Y803, G804, C805, L852, T862, D863, F864.
Sildenafil	L726, F731, V734, A751, K753, S783, L785, L796, V797, T798, Q799, L800, M801, Y803, G804, C805, L852, T862, D863, F864.
Terazosin	L726, F731, V734, A751, I752, K753, M774, S783, R784, L785, L796, V797, T798, Q799, L800, M801, Y803, G804, C805, L852, T862, D863, F864.
Topotecan	L726, F731, V734, A751, K753, S783, T798, Q799, L800, M801, Y803, G804, C805, D808, L852, T862, D863.
Trimetrexate	F731, V734, A751, I752, K753, M774, S783, R784, L785, L796, V797, T798, M801, Y803, G804, C805, L852, T862, D863, F864.

**Table 2 pharmaceuticals-14-00791-t002:** Interactions between inhibitors selected in DrugBank and EGFR during simulations.

Drug	Residues
Alfuzosin	L718, V726, A743, I744, K745, M766, C775, L777, L788, I789, T790, L792, M793, G796, C797, D800, R841, L844, N842, T854, D855, F856, L858.
Amodiaquine	L718, G719, V726, A743, Q791, L792, M793, P794, F795, G796, C797, D800, L844.
Antrafenine	L718, G719, S720, G721, G724, F723, T725, V726, K745, E746, L747, R803, L858, K913.
Bopindolol	L718, G719, V726, A743, I744, K745, L777, L788, I789, T790, L792, G796, C797, R841, N842, L844, T854, D855.
Carvedilol	L718, G719, S720, F723, T725, V726, K745, D746, L747, L788, D837, R841, N842, D855, L858, L862.
Doxasozin	L718, V726, A743, K745, Q791, L792, M793, P794, F795, G796, Y801, E804, H805, L844.
Pindolol	L718, S720, G721, V726, A743, K745, T790, Q791, L792, M793, G796, C797, R841, N842, L844, T854, D855.
Prazosin	L718, V726, A743, K745, M766, V769, C775, R776, L777, L788, T790, Q791, L792, M793, G796, C797, L844, T854, D855, F856.
Quinacrine	L718, G719, V726, A743, K745, T790, Q791, L792, M793, G796, C797, D800, R803, R841, L844, T854, D855.
Saprisartan	L718, G719, S720, G721, F723, G724, T725, V726, A743, K745, L747, T790, Q791, L792, M793, G796, C797, R841, L844, T854, L858, L862, K875, P877.
Terazosin	L718, G719, V726, A743, K745, C775, R776, M766, L777, L788, T790, M793, G796, C797, D800, L844, T854, D855, F856.
Topotecan	L718, G719, S720, G721, V726, A743, L792, M793, P794, G796, C797, D800, R841, N842, L844, D855.
Trimetrexate	L718, G721, A722, F723, V726, A743, K745, T790, Q791, L792, M793, G796, C797, R841, N842, L844, T854, D855, L858.
Udenafil	V717, L718, A743, Q791, L792, M793, P794, F795, G796, C797, L799, D800, R803, R841, L844.
Vardenafil	L718, G719, S720, G721, F723, G724, T725, V726, K745, E746, L747, L788, L792, G796, C797, D800, R841, N842, D855, L858.

**Table 3 pharmaceuticals-14-00791-t003:** Binding affinity for the protein-ligand systems (in units of kcal/mol).

System	ΔE_vdw_	ΔE_ele_	ΔG_ele,sol_	ΔG_npol,sol_	ΔG_mmgbsa_
HER2_alfuzosin_	−57.0 ± 4	−33.2 ± 6	45.4 ± 4	−7.1 ± 0.3	−51.9 ± 6
HER2_antrafenine_	−66.3 ± 3	24.4 ± 8	−0.9 ± 0.1	−8.3 ± 0.2	−51.1 ± 3
HER2_bopindolol_	−40.0 ± 4	18.7 ± 10	−7.8 ± 1	−5.5 ± 0.5	−34.6 ± 5
HER2_carvedilol_	−51.1 ± 6	16.0 ± 4	4.6 ± 1	−6.8 ± 0.6	−37.3 ± 5
HER2_doxasozin_	−49.0 ± 4	24.0 ± 9	−7.4 ± 1	−6.0 ± 0.3	−38.4 ± 4
HER2_irinotecan_	−68.0 ± 4	40.2 ± 12	−21.0 ± 8	−7.6 ± 0.3	−56.4 ± 5
HER2_pindolol_	−26.0 ± 3	4.0 ± 0.5	3.4 ± 0.6	−3.8 ± 0.4	−22.4 ± 4
HER2_prazosin_	−56.0 ± 3	−24.5 ± 7	42.7 ± 5	−6.9 ± 0.2	−44.7 ± 4
HER2_quinacrine_	−57.5 ± 3	34.0 ± 8	−24.0 ± 7	−7.4 ± 0.2	−54.9 ± 3
HER2_saprisartan_	−57.0 ± 4	−17.8 ± 5	46.3 ± 4	−7.2 ± 0.4	−35.7 ± 4
HER2_sildenafil_	−49.1 ± 3	34.7 ± 9	−22.7 ± 9	−6.2 ± 0.3	−43.3 ± 3
HER2_terazosin_	−46.0 ± 4	−38.3 ± 11	50.0 ± 7	−5.7 ± 0.4	−40.0 ± 7
HER2_topotecan_	−45.1 ± 3	−26.3 ± 4	43.0 ± 12	−5.3 ± 0.4	−33.7 ± 5
HER2_trimetrexate_	−51.6 ± 3	−23.6 ± 4	33.9 ± 3	−6.5 ± 0.3	−47.8 ± 4
EGFR_alfuzosin_	−55.5 ± 4	−30.3 ± 6	47.4 ± 5	−7.0 ± 0.3	−45.4 ± 4
EGFR_amodiaquine_	−30.8 ± 4	43.3 ± 20	−35.2 ± 15	−4.1 ± 0.6	−26.8 ± 8
EGFR_antrafenine_	−32.6 ± 4	65.4 ± 10	−48.5 ± 10	−4.0 ± 0.6	−19.7 ± 3
EGFR_bopindolol_	−40.6 ± 4	17.7 ± 9	−8.4 ± 1	−5.8 ± 0.4	−37.1 ± 3
EGFR_carvedilol_	−35.8 ± 5	−21.4 ± 11	25.8 ± 7	−5.4 ± 0.5	−36.8 ± 7
EGFR_doxasozin_	−38.8 ± 4	3.2 ± 0.9	9.7 ± 3	−4.6 ± 0.4	−30.5 ± 4
EGFR_pindolol_	−25.4 ± 5	4.7 ± 0.5	−4.1 ± 1	−4.4 ± 0.7	−29.2 ± 5
EGFR_prazosin_	−52.8 ± 3	−19.0 ± 6	38.2 ± 4	−6.6 ± 0.2	−40.2 ± 4
EGFR_quinacrine_	−40.6 ± 4	12.9 ± 3	−4.7 ± 0.5	−5.4 ± 0.5	−37.8 ± 4
EGFR_saprisartan_	−46.4 ± 3	−15.3 ± 2	46.2 ± 10	−6.4 ± 0.5	−21.9 ± 5
EGFR_terazosin_	−55.8 ± 4	−22.7 ± 7	40.4 ± 6	−6.6 ± 0.3	−44.7 ± 6
EGFR_topotecan_	−35.2 ± 7	−6.3 ± 1	22.7 ± 9	−3.9 ± 0.8	−22.7 ± 7
EGFR_trimetrexate_	−42.6 ± 3	−29.8 ± 7	48.5 ± 6	−5.6 ± 0.4	−29.5 ± 4
EGFR_udenafil_	−40.3 ± 3	10.0 ± 2	5.6 ± 0.5	−5.0 ± 0.3	−29.7 ± 3
EGFR_vardenafil_	−53.6 ± 5	−35.0 ± 12	46.8 ± 11	−6.4 ± 0.7	−48.2 ± 7

**Table 4 pharmaceuticals-14-00791-t004:** Per-residue free energy for complexes formed between alfuzosin, antrafenine, irinotecan, and quinacrine with HER2 (values kcal/mol).

Residue	HER2_alfuzosin_	HER2_antrafenine_	HER2_irinotecan_	HER2_quinacrine_
L726	−1.0	−1.1	−2.6	−1.8
G727		−1.0	−0.9	
F731	−1.1		−1.9	−1.6
T733		−0.7		
V734	−1.6	−3.0	−2.6	−2.1
A751	−0.9	−1.0	−1.6	−1.1
K753	−1.3	−1.2	−1.7	−1.5
M774	−1.1			
S783	−0.6		−0.5	−0.7
L785	−1.8	−0.9	−1.0	−0.9
L796	−1.4	−1.9	−0.5	
V797	−0.5	−0.9	−1.0	
T798	−1.8	−1.8	−1.8	−5.7
L800				−0.7
M801	−0.9	−1.8		
P802			−1.7	−1.0
Y803			−1.7	−2.0
G804				−0.6
C805	−0.6	−0.6	−0.5	−0.6
L852	−1.8	−1.4	−1.3	−1.9
T862	−1.1	−1.2	−1.1	−2.0
D863	−1.9	−0.7		
F864	−1.6		−0.5	

**Table 5 pharmaceuticals-14-00791-t005:** Per-residue free energy for complexes formed between alfuzosin, prazosin, terazosin, and vardenafil with EGFR (values kcal/mol).

Residue	EGFR_alfuzosin_	EGFR_prazosin_	EGFR_terazosin_	EGFR_vardenafil_
L718	−2.2	−1.2	−2.2	−2.4
G719				−0.9
G721				−1.2
F723				−0.6
T725				−0.7
V726	−1.9	−2.0	−2.1	−3.0
A743	−1.0	−0.9	−1.0	−0.7
I744			−0.7	
K745	−1.4	−1.2	−2.0	−3.2
M766			−0.6	
L777		−1.9	−1.1	
L788	−1.0	−1.1	−1.6	
I789	−1.3		−0.5	
T790	−0.5	−1.2	−1.1	
L792		−0.9	−0.8	−0.9
M793	−1.0	−1.1	−0.7	−0.5
G796	−1.0	−0.5	−1.0	−0.7
C797	−1.0	−0.4	−0.9	−1.3
R841				−2.0
L844	−2.0	−2.2	−2.2	−0.8
T854	−1.3	−1.7	−1.7	
F856	−1.0	−0.6	−0.5	
L858	−0.6			−0.6

**Table 6 pharmaceuticals-14-00791-t006:** Cytotoxic activity (IC_50_ values in µM *) in two breast cancer cell lines at 48 h of incubation.

Compounds	MCF-7	MDA-MB-231
Gefitinib	10 ± 1	13 ± 1
Lapatinib	7 ± 1	10 ± 1
Terazosin hydrochloride	70 ± 1	68 ± 2
Alfuzosin hydrochloride	85 ± 2	69 ± 1
Prazosin hydrochloride	41 ± 2	25 ± 1
Irinotecan hydrochloride	0.37 ± 0.8	27 ± 1
Quinacrine dihydrochloride	1.4 ± 1	1.3 ± 1

* The calculation of Inhibitory concentration 50 (IC_50_) of the compounds was determined by the cell viability percentage curve vs the logarithmic of concentration.

## Data Availability

Not applicable.

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
