# Peer review of "Repurposing FDA Drug Compounds against Breast Cancer by Targeting EGFR/HER2"

_pharmaceuticals, 2021, doi:10.3390/ph14080791_

Round 1

Reviewer 1 Report

This is a well written paper on repurposing studies by screening DrugBank in the search for new FDA-approved drugs with chemical structures similar to lapatinib and gefitinib.

Is work relevant/novel?- novelty to be highlighted in the article. 
Please highlight how this study adds to the current knowledge, as mentioned in the paper, resistance develops easily for lapatinib
How was the Databank used? Please explain more in the methodology
What are the limitations of this study

Author Response

Rewiever 1

This is a well written paper on repurposing studies by screening DrugBank in the search for new FDA-approved drugs with chemical structures similar to lapatinib and gefitinib.

Is work relevant/novel?- novelty to be highlighted in the article. 

Response:

The novelty of the article has been highlighted in the manuscript (page 15).

Please highlight how this study adds to the current knowledge, as mentioned in the paper, resistance develops easily for lapatinib

Response:

The requested information has been added in the manuscript (page 15).

How was the Databank used? Please explain more in the methodology
What are the limitations of this study

Response:

More information about the inclusion criteria for the choice of compounds have been added to the manuscript (page 13).

Reviewer 2 Report

The manuscript entitled "Repurposing FDA drug compounds against breast cancer by targeting EGFR/HER2" proposes the use of other compounds, differents from the normally used, as therapeutic targets of EGFR / HER-2 in breast cancer. The authors show that there are other compounds that are more effective than those used so far.
The study is of scientific interest and the proposed work is well written.
However, the results obtained are so promising that they should further reinforce the discussion of them, since it remains a bit inconsistent.
On the other hand I would only have one question:
Did the authors take into account any specific inclusion criteria for the choice of compounds? It should be specified in material and methods section.

Author Response

Rewiever 2

The manuscript entitled "Repurposing FDA drug compounds against breast cancer by targeting EGFR/HER2" proposes the use of other compounds, differents from the normally used, as therapeutic targets of EGFR / HER-2 in breast cancer. The authors show that there are other compounds that are more effective than those used so far.
The study is of scientific interest and the proposed work is well written.
However, the results obtained are so promising that they should further reinforce the discussion of them, since it remains a bit inconsistent.
On the other hand I would only have one question:
Did the authors take into account any specific inclusion criteria for the choice of compounds? It should be specified in material and methods section.

Response:

More information about the inclusión criteria for the choice of compounds have been added to the manuscript (page 13).

Round 2

Reviewer 1 Report

Thank you for the revisions